# Multi-Asymmetric Irradiation Method Using a Ring Array to Obtain an Emission-Capable LED Beam Power Effect to Observe Cancer Removal Status in a Surgical Microscope

**DOI:** 10.3390/diagnostics13223482

**Published:** 2023-11-20

**Authors:** Seon Min Lee, Kicheol Yoon, Sangyun Lee, Seung Yeob Ryu, Kwang Gi Kim

**Affiliations:** 1Department of Nursing, College of Nursing, 191 Hambakmoero, Yeonsu-gu, Incheon 21936, Republic of Korea; 990727leesm@gmail.com; 2Medical Devices R&D Center, Gachon University Gil Medical Center, 21, 774 Beon-gil, Namdong-daero Namdong-gu, Incheon 21565, Republic of Korea; kcyoon98@gachon.ac.kr (K.Y.); l0421h@gmail.com (S.L.); wyverns0723@gmail.com (S.Y.R.); 3Department of Biomedical Engineering, College of Health Science & Medicine, Gachon University, 1342 Seongnamdaero, Sujeong-gu, Seongnam-si 13120, Gyeonggi-do, Republic of Korea; 4Department of Biohealth & Medical Engineering Major, Gachon University, 1342 Seongnamdaero, Sujeong-gu, Seongnam-si 13120, Gyeonggi-do, Republic of Korea; 5Department of Health Sciences and Technology, Gachon Advanced Institute for Health Sciences and Technology (GAIHST), Gachon University, 155 Gaetbeol-ro, Yeonsu-gu, Incheon 21999, Republic of Korea

**Keywords:** asymmetrical LED beam irradiation, increasing of beam width, increasing of beam power, fluorescence emission-guided, surgical microscope

## Abstract

The light emitting diodes (LEDs) used in surgical fluorescence microscopes have weak power, to induce fluorescence emission. The LED induces fluorescence emission throughout a lesion due to its large beam width; however, the beam irradiation intensity is not uniform within the beam width, resulting in a fluorescence emission induction difference. To overcome this problem, this study proposes an asymmetric irradiation array for supplying power uniformly throughout the beam width of the LED and increasing the intensity of the LED. To increase the irradiation power of the LEDs, a multi-asymmetric irradiation method with a ring-type array structure was used. The LED consisted of eight rings, and the space between the LEDs, the placement position, and the placement angle were analyzed to devise an experimental method using 3D printing technology. To test the irradiation power of the LED, the working distance (WD) between the LED and target was 30 cm. The bias voltage of the LED for irradiating the light source was 5.0 V and the measured power was 4.63 mW. The brightness (lux) was 1153 lx. Consequently, the LED satisfied the fluorescence emission induction conditions. The diameter of the LED-irradiated area was 9.5 cm. Therefore, this LED could be used to observe fluorescent emission-guided lesions. This study maximized the advantages of LEDs with optimal conditions for fluorescence emission by increasing the beam width, irradiation area, and energy efficiency, using a small number of LEDs at the maximum WD. The proposed method, optimized for fluorescence expression-induced surgery, can be made available at clinical sites by mass producing them through semiconductor processes.

## 1. Introduction

Malignant tumors with high infiltration levels rapidly metastasize and have a high risk of recurrence [1,2,3]. Surgeries aim for the complete resection of malignant tumors [1,2,3]. However, tumors may be distributed in blood vessels [4].

The colors of the tumor and blood vessels are similar, making it difficult to visually observe the boundaries [5]. Therefore, the state of tumor removal can be observed using a fluorescence surgical microscope by injecting fluorescence sodium (Fluorescite injection 10% [500 mg] @ Korea Alcon. Co., Ltd., Seoul, Republic of Korea) into the body [6,7,8,9]. In this study, a light emitting diode (LED), suitable for use as a fluorescent contrast agent, was irradiated in vitro. When an LED with a 405 nm excitation wavelength (λ_ext_) was irradiated on the tumor, a 530–560 nm fluorescence emission wavelength (λ_em_) was generated by a chemical reaction of the fluorescent contrast agent present in the tumor [6]. The color (yellow), corresponding to the spectrum of fluorescence emission, was photographed using a camera. Finally, the captured images were observed using an external monitor.

The fluorescent contrast medium must have sufficient LED beam strength (>0.5 mW) and beam width (30–60°). If the beam width is narrow, the lesion observation field becomes narrow [10]. For observation with a clear image quality, the working distance (WD) between the LED (or camera) and the lesion must be 15–30 cm, and the range (θ_D_) of the left and right movement of the LED (or camera) must also correspond to 30–60° [10]. However, a WD of 15–30 cm consumes LED beam energy for delivery to the lesion, resulting in LED power loss [11,12]. Therefore, the fluorescence emission of small lesions may be weak. In addition, surgical microscopes are large and heavy, with high maintenance costs [13]. Therefore, it is necessary to develop a handheld lightweight microscope, with a free radius of movement of the WD and cameras (or LEDs) to make lesion observation easier. The beam energy loss generated by WD is small. Commercial LEDs are utilized to minimize the size and cost. However, it is impossible to secure the lesion observation field due to the narrow beam width (θ_D_ = 10°) of LEDs [14]. That is, the beam width of LEDs is wide, and the observation field is irradiated more uniformly as the beam width increases. Therefore, lesions > 10 cm in size can be observed. The number of LEDs can be increased to increase the beam width and irradiation intensity. However, LED irradiation methods lead to the formation of shadows within the lesion, and the costs increase with a rising number of LEDs.

Shadows interfere with the visualization of the lesion. Therefore, it is necessary to increase the beam strength and width by using only a small number of LEDs. As WD increases, illuminance decreases and the intensity of the light is not uniform, even within the beam width [14]. Blue LEDs are highly utilized due to their small volume and low cost, but because the beam width (θ_D_ = 10°) is relatively narrow (diameter of survey area = 26 mm), it is difficult to secure a lesion viewing angle. As this leads to a decrease in the accuracy of lesion observation and diagnosis using a surgical microscope, research is needed to design an LED arrangement that expands the beam width of light and minimizes shadow formation in the lesion.

LED arrangements in the form of rings and array matrices have been studied and developed [15,16]. These arrangements are suitable for fluorescence emission-guided sample monitoring in the operating room because of their uniform light intensities and sufficient observation fields of the object. The ring-type LED was designed with a wide-wavelength band light source by placing RGB LEDs at the same ratios as those for application in machine vision systems. However, considering the nature of the wavelength of LEDS with applications in surgical microscopes, the fluorescence emission-guided LED irradiation wavelength is a single wavelength, and is therefore not suitable for sample observation [15]. LEDs arranged in an array matrix exceed the power level required for surgery [16]. These have the disadvantage of being burdened by applications in operating rooms because of their unnecessary energy consumption and high manufacturing costs, and it is difficult to adjust the radius angle and WD of the light source.

This paper proposes a method for designing ring-type asymmetric LED beam irradiation with a wide beamwidth, based on the uniform beam irradiation intensity of an LED, to secure a field of view so that lesions with large areas can be sufficiently observed through fluorescence emission. The proposed ring-type asymmetric LED beam irradiation method can optimize the number of LEDs, minimize beam irradiation power loss at a fixed WD, and secure a sufficient observation field and beam irradiation strength that can induce fluorescence emission.

## 2. Light Source Irradiation Characteristics Analysis and LED Arrangement Method

As shown in Figure 1a, LED beam irradiation methods are generally classified as symmetric or asymmetric beam irradiation. As shown in Figure 1b, there is a difference in the incident angles (θ_ext_) on the irradiation area (t) of the target [17].

A typical LED has an incident angle (θ_ext_) of 0°. Therefore, when light radiates from an LED, the beamwidth of the light is constant (θ_D_ = 10°), and the light spreads evenly in all directions. These methods are classified as symmetric beam irradiation techniques. If the LED is tilted away from 0°, the beam spreads out. Therefore, the beamwidth is no longer 10°. This is known as the asymmetric method.

The symmetric method has a relatively small beamwidth but a strong beam intensity. In contrast, the asymmetric method has a wide beamwidth but a weak beam intensity. The common point between the two methods is that the beam intensity is strong when the beam is perpendicular to the irradiation area of the LED. The longer the working distance, the weaker the beam intensity becomes. However, when the beam deviates from the perpendicular direction, its intensity (*P_h_* ≠ *P_l_*) weakens.

The dual asymmetric method can widen the beam and increase its intensity, as shown in Figure 1a. However, as the intensity of the beam becomes stronger only in the area where the LED beamwidths overlap, a shaded area forms, obstructing the view of the lesion. The multi-asymmetric method widens the beam and uniformly increases its intensity. Although they can provide the most optimal effect, dual- and multi-asymmetric methods increase the number of LEDs, resulting in increased costs and complicated structures.

A ring-type array can increase the number of LEDs and widen the beam width, as shown in Figure 2. In addition, the power and brightness of the beam can be increased by using more LEDs. Because of these functional features, a ring-type array design can be used to effectively observe the target, as shown in Figure 1b. A structural feature of the ring-type array is that it is possible to install a camera in the center hole (h), which can reduce the formation of shadows in the monitored image [18].

LEDs (Thorlabs LED405E) with a beamwidth (θ_D_) of 10° are attached in a ring-type array structure, maintaining a constant distance (D_r_: 4.32 cm) between the LEDs, as shown in Figure 2. As shown in Figure 1b, when collectively irradiating the target (lesion), there might be areas where the light is unable to reach uniformly. Therefore, shadow generation must be eliminated to secure the field of view for fluorescence emission observations. The symmetric irradiation method has limitations in securing a uniform observation field of the lesion, due to the formation of shadows. In addition, it is uneconomical because the standards of the light source, the manufacturing cost, and the power consumption increase with an increase in the number of LEDs. Therefore, it is necessary to propose an LED beam irradiation method in which the entire lesion can emit fluorescence using only a small number of LEDs without shadow formation. An asymmetric beam irradiation method rotates the incident angle (θ_ext_ > 0°) so that the irradiation area of light becomes wider than the symmetric irradiation area, as shown in Figure 1. The single asymmetric irradiation method has different excitation powers (*P_h_* ≠ *P_l_*) in both directions at the center point (d_g_) of the irradiation area (t); therefore, the fluorescence is not uniform within the irradiation area (t), and the light source generates a different power level for each position of the beam width. To compensate for this phenomenon, we propose a multi-asymmetric beam irradiation method. The multi-asymmetric beam irradiation method increases the number of LEDs, as shown in Figure 2. The LEDs are arranged with a relatively weak power overlap. Therefore, the proposed method can induce uniform fluorescence (*P_h_* = *P_l_*) and is designed to enable fluorescence emission in a wider area. It has the advantage that fewer LEDs are used, compared to a symmetric beam irradiation method under an identical WD, light source size, and irradiation power condition.

When performing LED beam irradiation, the variables of LED beam irradiation are the number of LEDs, the arrangement interval (D_r_), the incident angle (θ_ext_), and the WD. To ensure sufficient resolution, even at the maximum WD, the maximum WD of the multi-asymmetric beam irradiation method was determined to be 30 cm. To obtain the optimal angle of incidence for LED irradiation, it was necessary to observe the change in the irradiation area of the LED as the angle of incidence changes, while the other performance variables are fixed. Therefore, the number of LEDs and the arrangement spacing should be identical, as shown in Figure 2. The LED irradiation area (t) increases in proportion with the incident angle (θ_ext_), as shown in Figure 1. Shadows form when the incident angle (θ_ext_) is more than 15.37°. Figure 3a shows that, if the angle of incidence (θ_ext_) increases for the irradiation sections of LED_n_ and LED_n+4_, the width of the irradiation area from LED_n_ to LED_n+4_ increases proportionally, and the shadow phenomenon occurs between each irradiation area (t). When observing a lesion during surgery, shadows always occur at the center of the lesion. Therefore, the shaded area is not irradiated with an LED (at the excitation wavelength). Therefore, fluorescence emission (at the emission wavelength) cannot occur in shaded areas, and lesions cannot be observed in these areas. Additionally, if a lesion is observed with the brightness of the LED, the shaded area is not irradiated by the LED, which obstructs the view during the lesion observation process. Quantification of the irradiated area is required to evenly irradiate the entire area of the lesion with an LED. As shown in Figure 3b, if eight LEDs simultaneously irradiate the entire area (A_ext_) of the lesion, depending on the incident angle (θ_ext_) given in Figure 3a, the single LED irradiation areas (A_s_) might overlap, and if not, shaded areas might occur. As the result, with the boundaries of the overlapping regions canceled out, each irradiation area (A_s_) remains in the form of the sum of areas (A_ext_). Each irradiation area (A_s_) has an elliptical shape. This is because LED irradiation is asymmetric. Therefore, it is necessary to obtain the value of the irradiation area (A_ext_) of the LED, to obtain results for the entire area of the lesion to be irradiated.

As a condition for the test, a minimum incident angle (θ_ext_), capable of irradiating the entire area of the phantom, must be established. As shown in Figure 4a, during the design process of the experiment, the fixing frame of the LED was manufactured using a 3D printer. The incident angle (θ_ext_) of the LED is 15.37°, as shown in Figure 4b.

## 3. Experimental Results

To test the light source performance of the proposed method, a phantom made of liquid latex rubber was injected with a fluorescent contrast agent of sodium fluorescein (Alcon, Seoul, Republic of Korea, 5 mL of Fluorescite injection 10%) at a concentration of 0.02 mM [19]. This phantom was used to represent lesions. The size of the phantom was approximately 10 cm, which is similar to that of colon cancer lesions. Referring to Figure 2, the power reaching the phantom (P_r_) and the diameter of the irradiance area were measured. The incident angle and beam irradiation power (P_ext_) of all LEDs were 15.37° and 10 mW, respectively, at an irradiation wavelength of 405 nm, and the total number of LEDs was eight.

The LEDs were arranged in a ring-array type structure. The diameter of the beam irradiation area for the guided fluorescence emission was 9.5 cm. This is sufficient to irradiate most types of tumor tissue. To set up the experimental conditions, the light source was fixed to a clamp (Manfrotto 244N, Manfrotto, Veneto, Italy), as shown in Figure 5, and the height of the clamp was 30 cm. Thus, the working distance between the LED and target was 30 cm.

The bias voltage of the LED for irradiating the light source was 5.0 V. Light power (P_r_) and light brightness were measured using a power meter (Thorlabs S121C) and a lux meter (Thorlabs S142C, Sincon ST-126), respectively. The irradiation area of the light source was measured, as shown in Figure 5. The measurement results are presented in Table 1.

The measured power (P_r_) of the light source was 4.63 mW at an irradiation wavelength of 405 nm, and this power satisfied the conditions for fluorescence emission (>0.5 mW). These fluorescence results are shown in Figure 6 and were observed using an NIR camera (Lt-225c, Lumenera camera, TELEDYNE, Inc., Thousand Oaks, CA, USA) and a phantom to induce fluorescence. With a WD of 30 cm, the LED supply bias voltage was adjusted from 3.0 to 5.0 V, and the phantom area of interest (ROI) in the NIR camera image was converted into an average RGB value.

The irradiance presented in Figure 6 can be calculated using the average RGB value, which means that the higher the brightness value, the better the fluorescence emission. The proposed light source emits fluorescence at the maximum WD (30 cm).

The irradiation area of the LED decreased proportionally with a decrease in the WD. Therefore, a WD of at least 15 cm was used to irradiate the entire phantom. This phenomenon was observed at the maximum LED beam irradiation intensity and an external voltage of 3.5 V or higher. As shown in Figure 6, the light source has excellent light power, brightness, and fluorescence emission. The phantom area (A_ext_) where fluorescence emission occurred under the irradiation of the light source was measured to be 9.5 cm.

## 4. Discussion

A 405 nm LED with a beam width (θ_ext_) of 10° and an optical source power of 10 mW was used to analyze fluorescence emission. The experiment was conducted after injecting a fluorescent contrast agent into the phantom [19]. Most surgeries using 405 nm fluorescein sodium are craniotomies [20]. Compared to the incision size of the craniotomy, the irradiation area of the light source was suitable (diameter of 9.5 cm). Therefore, the fluorescence emission-guided multi-asymmetric beam irradiation method using sodium fluorescein is suitable for observing tumor removal during surgery. In a symmetric LED irradiation method, direct irradiation must be performed in a straight line using a beam with a 0° reference (Figure 1). Therefore, only the LED-irradiated position of the lesion becomes stronger with increasing LED intensity. In addition, because the beams are not irradiated at intervals between LEDs, a shadowed area may form. Table 2 summarizes the differences between the proposed method and other research results. It is difficult to obtain an absolute comparison with other studies ([15,16,21,22]) because the WD, LED quantity, and irradiation power were all different. The power of the LED depends on the number of LEDs, and the WD depends on the research objectives. Therefore, we were interested in the received power and irradiation area, to determine whether the fluorescence emission was sufficiently wide (wide irradiation area) and strongly irradiated for a WD of 30 cm, when applied to a surgical field using a small number of LEDs. We compared our results with those in references [15,16,21,22]. The number of LEDs was small compared to those in [15,16,21,22], and the WD was longer. The beam irradiation power was relatively low. The received power and irradiation area were the largest.

The prior studies share a common design feature; a dome-type array structure [21,22]. The dome-type array structures were connected by using 90 and 130 LEDs, respectively [21,22]. The reason for increasing the number of LEDs is that the brightness of the light source increases proportionally with the number of LEDs. Thus, even if the WD increases, the irradiation brightness of the LEDs and the irradiation beam width increase proportionally, and a large area can be irradiated uniformly. However, the optimal WD range at a surgical site is 15–30 cm [10]. Therefore, an excessive WD increase is unnecessary. If the WD increases, the intensity of the irradiation beam should be increased by increasing the number of LEDs, so that fluorescence can be sufficiently expressed. However, an increase in the number of LEDs and the intensity of beam irradiation increases the energy consumption, which causes heat generation in the LED module and mechanical damage to the module [22]. The studies adjusted the excitation angle range from 0 to 65° to increase the beam irradiation effect [21,22]. In particular, the incident angles were adjusted differently for each LED layer (θ_0_ = 47.5°, θ_1_ = 42.0°, θ_2_ = 36.6°, θ_3_ = 31.0°, θ_4_ = 22.9°) to enable a wide area of beam irradiation [22]. However, the actual irradiated area is not the only relevant factor for observing tumors during fluorescence-induced surgery. Some other factors include the angle of incidence, the WD, and the arrangement, direction, and power of LED.

For a surgical fluorescence induction microscope, with eight LEDs and a WD of 30 cm, the LED beam irradiation was strong enough to obtain an optimal received power (>0.05 mW) for fluorescence expression. This study analyzed an asymmetric LED beam irradiation method to irradiate a wide lesion with uniform intensity. This resulted in an even expression, making it is easier to observe the entire lesion. This was achieved by using a ring-type array design for LED beam irradiation.

## 5. Conclusions

This study contributes to the optimization of the angle, direction, ring diameter, and number of LEDs for LED beam irradiation. An LED beam can irradiate a wide area with uniform intensity. This analysis yielded structural and orientation angles with high confidence. The configuration of the experimental environment for testing used computer-aided design (CAD) drawing work and 3D printing technology, because of the precise angle, direction, ring diameter, and optimization of the number of LEDs required for beam irradiation. This design method enables the observation of lesions without shadows through fluorescence emission because of its uniform beam irradiation intensity and wide irradiation width. An experimental environment was constructed to obtain the results using these analyses and techniques, and positive results were obtained through photodynamic experiments. In surgical procedures for cancer or lymph node removals, we envision a technical method that involves the injection of a fluorescent contrast agent into the body and subsequent irradiation with an external light source to observe the lesion. A technical solution was developed to improve the irradiation width and intensity of the light source. This way, a sufficient observation field was secured to observe wide lesions using a light source and fluorescence emission. Securing the field of vision for lesion observation can improve the accuracy of observations. The proposed method uniformly increases the intensity of the beam by adjusting the irradiation angle of a small number of LED-based beams and secures a wide irradiation area while applying the asymmetric LED design method. This methodology can be applied in the surgical field for observing the status of tumor removals, blood circulation, and lymph nodes.

## Figures and Tables

**Figure 1 diagnostics-13-03482-f001:**
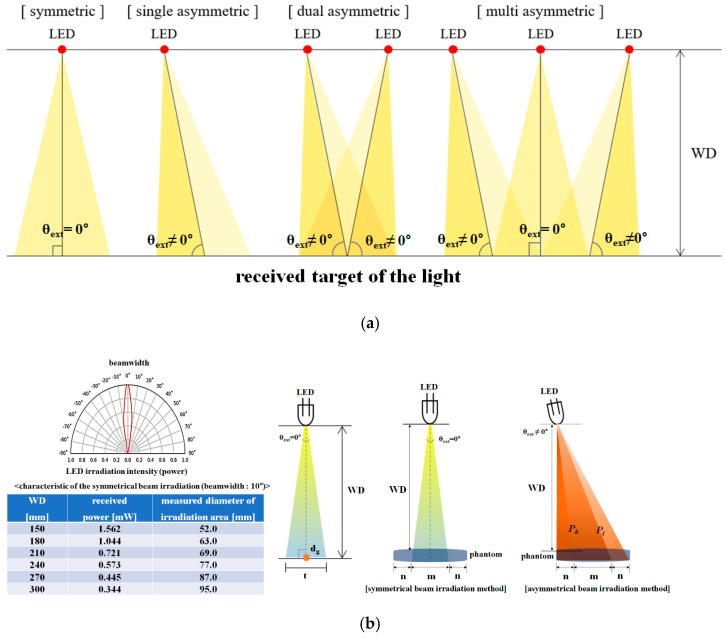
Characteristics of LED beam irradiation: (**a**) the difference between symmetric and asymmetric LED irradiation; (**b**) LED beam power and width.

**Figure 2 diagnostics-13-03482-f002:**
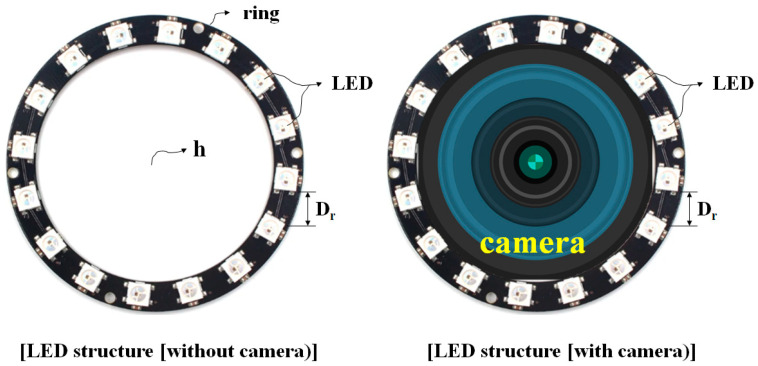
Structure of a symmetrical ring-type array LED.

**Figure 3 diagnostics-13-03482-f003:**
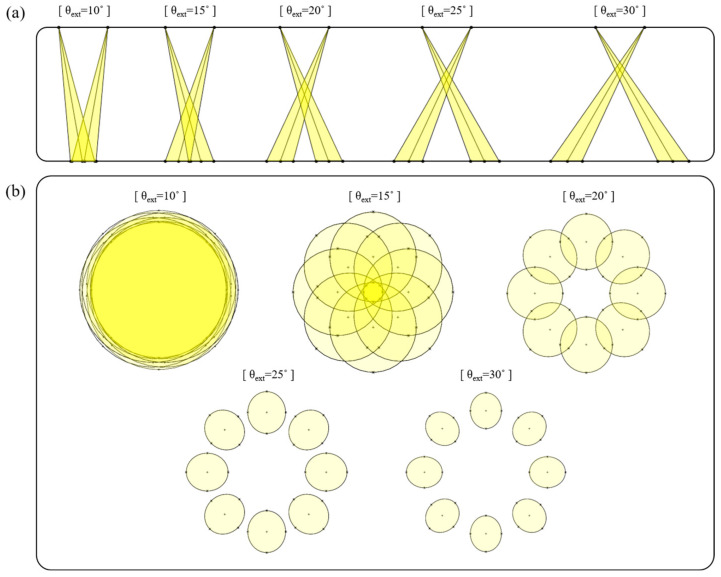
Analysis of LED irradiation. (**a**) Longitudinal section of LED_n_ and LED_n+4_ (10° ≤ θ_ext_ ≤ 30°), (**b**) bottom horizontal section of all LED irradiated (10° ≤ θ_ext_ ≤ 30°).

**Figure 4 diagnostics-13-03482-f004:**
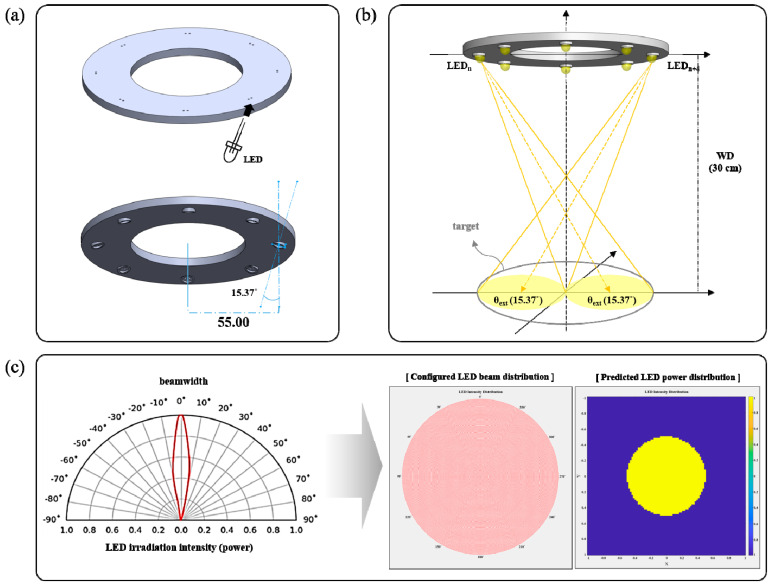
A schematic of the proposed light source: (**a**) structure, (**b**) beam shape during LED irradiation, (**c**) simulation results of P_r_ (received power) and beam distribution.

**Figure 5 diagnostics-13-03482-f005:**
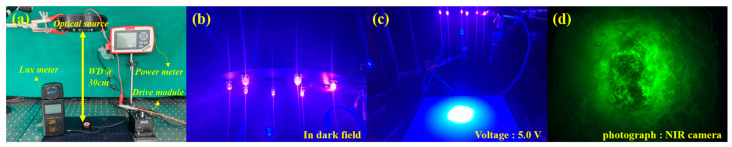
(**a**) Experiment environment, (**b**) image with light source on, (**c**) real irradiation pattern, and (**d**) image taken with phantom/syringe NIR camera.

**Figure 6 diagnostics-13-03482-f006:**
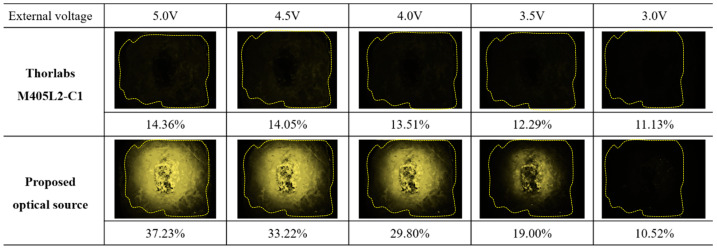
Comparison of the fluorescence emission of the proposed LED and a conventional LED.

**Table 1 diagnostics-13-03482-t001:** Electrical performance of LEDs.

Performance	Parameter	Unit
External voltage	5.0	V
Incidence angle	15.37	deg
Diameter of irradiance area	9.5	cm
Received power of center point	4.63	mW
Illuminance (lux)	1153	lx

**Table 2 diagnostics-13-03482-t002:** Comparison of performances of proposed and other LEDs.

Ref. [#]	λ_ext_ [nm]	WD [cm]	P_max_ [mW]	LED Quantity [pcs]	P_r_ [mW]	θ_ext_ [Degree]	Irradiance Area [cm^2^]	Characteristic
[6]	467	3.77	864	12	4.837	65	50.24	LED
[7]	550	56.75	7920	9	0.196	0	5.72	LED
[12]	748	50	581	90	0.018	12	12.25	LED
[13]	600–650	30	300,300	130	26.550	22.9	3.24	LED
This work	405	30	960	8	4.63	15.37	70.85	LED

## Data Availability

The data presented in this study are available upon request from the corresponding author. The data are not publicly available due to privacy and ethical restrictions.

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
