# Peer review of "Multi-Asymmetric Irradiation Method Using a Ring Array to Obtain an Emission-Capable LED Beam Power Effect to Observe Cancer Removal Status in a Surgical Microscope"

_diagnostics, 2023, doi:10.3390/diagnostics13223482_

Round 1
Reviewer 1 Report
Comments and Suggestions for Authors
By improving the wide beam width, irradiation area, and energy efficiency by utilizing a small number of LEDs at maximum WD, this work maximizes the benefits of having LEDs with optimal conditions for fluorescence emission. The paper presents an exciting and vital topic. However, I have some issues that need to be considered before publication:
1. Line 25, “Working distance” should be “working distance”.
2. Line 25, “30cm” to “30 cm”.
3. Line 45, “The color of the tumor and blood vessels is similar, making it difficult to visually 45 observe the boundary classification.” This information needs reference.
4. Line 207, “10mW” to “10 mW”
5. It is not clear what type of phantom was used. The chemical composition of the utilized phantom should be mentioned.
6. Line 247, “10mW” to “10 mW”
7. The number of references is 13. Appropriate scientific manuscripts should have at least 20 references and include recent references published in the last two years.
Comments on the Quality of English LanguageModerate editing of the English language is required. The repeated use of "in this case" and "in this time" is not appropriate. Kindly please revise the language of the manuscript, maybe with the help of a native English colleague.
Author Response
|
Comments 1: Line 25, “Working distance” should be “working distance”. |
|
Response 1: Thank you for pointing this out. We agree with this comment. Therefore, we have revised "Working distance" to "working distance". Please review the corrections highlighted yellow color in Line 27. |
|
Comments 2: Line 25, “30cm” to “30 cm”. |
|
Response 2: Agree. We have, accordingly, revised "30cm” to "30 cm”. Please review the corrections highlighted green color in Line 28. |
|
Comments 3: Line 45, “The color of the tumor and blood vessels is similar, making it difficult to visually 45 observe the boundary classification.” This information needs reference. |
|
Response 3: Agree. Therefore, we have appended a reference corresponding to this information [5]. Please review the corrections highlighted green color in line 43 to 44. |
|
Comments 4: Line 207, “10mW” to “10 mW” |
|
Response 4: Agree. We have, accordingly, revised "10mW” to "10 mW”. Please review the corrections highlighted red color in line 202. |
|
Comments 5: It is not clear what type of phantom was used. The chemical composition of the utilized phantom should be mentioned. |
|
Response 5: We agree with this comment. Therefore, we have specified the material and chemical composition of the phantom in line 196 to 198 with red color. |
|
Comments 6: Line 247, “10mW” to “10 mW” |
|
Response 6: Agree. Accordingly, we have revised "10mW” to "10 mW”. Please review the corrections highlighted red color in line 239. |
|
Comments 7: The number of references is 13. Appropriate scientific manuscripts should have at least 20 references and include recent references published in the last two years. |
|
Response 7: We have appended references and included references published within the last two years. Please refer to the references [1], [4], [10], and [14] and the introduction session. |
|
4. Response to Comments on the Quality of English Language |
|
Point 1: Moderate editing of the English language is required. The repeated use of "in this case" and "in this time" is not appropriate. Kindly please revise the language of the manuscript, maybe with the help of a native English colleague. |
|
Response 1: We agree with this comment. We have commissioned a professional review of the English for this manuscript. Thus, we have completely revised the manuscript to reflect the English revision and have attached a certificate. |
|
5. Additional clarifications |
|
We are again grateful to the reviewers for providing a detailed review. We have revised the manuscript based on the comments, of course, but we also upgraded the figures and other ambiguous phrases. We have attached the manuscript with all comments incorporated. |
Reviewer 2 Report
Comments and Suggestions for Authors
Reviewer’s comments on manuscript „Multi-asymmetric irradiation method of ring-array type to obtain an emission-capable LED bem power effect to observe cancer removal status in a surgical microscope (Diagnostics-#2595021)” by S.M. Lee et al.
This work is about the optimization of fluorescence aided LED array illuminations in surgical microscopes. Because of the importance meant by cancer removal, this work has high practical value and it is expected to find high interests of health professionals working in the surgical field. The illustrating modell experiments are well done and the specifications of the applied equipments are precisely given. My comments are mainly related with the quality of presentation of the results.
Subjectal concerns:
1. The asymmetrical beam irradiation mode is not clear, merely based on Fig. 1 Panel „b”. What is the explanation of asymmetry in this case? Another explaining figure is recommended.
2. Also on Fig. 1 Panel „a”, Qext should be 0°, isn’t it? Please correct it.
3. Please give in the text also the SI unit of „lux” at its first occurrence.
4. In Table 3 please detail the abbreviation „ea”. This is rather uncommon.
5. Please improve the visibility of the objects in Fig. 5 Panel „a”.
Formal concerns:
1. On Fig. 4, please increase the letter size. The shaded arrows are hard to reveale that these are really arrows. These should be elongated.
2. In the „Abstract”, please write „diameter of LED irradiation area (6.294 cm)”.
3. Overseeing the whole text, regarding style, by a native English speaker is strongly recommended. Additionally there are a lot of mis-spellings. Please correct them.
Comments on the Quality of English LanguageIt should be improved extensively.
Author Response
|
Comments 1: The asymmetrical beam irradiation mode is not clear, merely based on Fig. 1 Panel „b”. What is the explanation of asymmetry in this case? Another explaining figure is recommended. |
|
Response 1: We agree with this comment. Therefore, we have appended the explanations for Figure 1. Please review the corrections highlighted red color in the lines of 109 to 126. |
|
Comments 2: Also on Fig. 1 Panel „a”, Qext should be 0°, isn’t it? Please correct it. |
|
Response 2: Agree. We have, accordingly, revised the figure. Please refer to Figure 1(a). |
|
Comments 3: Please give in the text also the SI unit of „lux” at its first occurrence. |
|
Response 3: Agree. We have appended the SI unit of the lux. Please refer to the corrections highlighted red color in line 29. |
|
Comments 4: In Table 3 please detail the abbreviation „ea”. This is rather uncommon. |
|
Response 4: Agree. We have revised the “ea” to the SI unit of pcs. Please refer to the corrections highlighted red color of Table 3. |
|
Comments 5: Please improve the visibility of the objects in Fig. 5 Panel „a”. |
|
Response 5: Agree. We have altered the image to increase the clarity. Please refer to Figure 5(a). |
|
Comments 6: On Fig. 4, please increase the letter size. The shaded arrows are hard to reveale that these are really arrows. These should be elongated. |
|
Response 6: We have revised Figure 4 to increase the letter size and arrows. |
|
Comments 7: In the „Abstract”, please write „diameter of LED irradiation area (6.294 cm)”. |
|
Response 7: We have appended the diameter of LED irradiation area to the abstract. Please refer to the corrections highlighted red color in line 30. |
|
Comments 8: Overseeing the whole text, regarding style, by a native English speaker is strongly recommended. Additionally there are a lot of mis-spellings. Please correct them. |
|
Response 8: We agree with this comment. We have commissioned a professional review of the English for this manuscript. Thus, we have completely revised the manuscript to reflect the English revision and have attached a certificate. |
|
4. Response to Comments on the Quality of English Language |
|
Point 1: |
|
Response 1: |
|
5. Additional clarifications |
|
We are again grateful to the reviewers for providing a detailed review. We have revised the manuscript based on the comments, of course, but we also upgraded the figures and other ambiguous phrases. We have attached the manuscript with all comments incorporated. |